# Neuronal Circuit Dysfunction in Amyotrophic Lateral Sclerosis

**DOI:** 10.3390/cells13100792

**Published:** 2024-05-07

**Authors:** Andrea Salzinger, Vidya Ramesh, Shreya Das Sharma, Siddharthan Chandran, Bhuvaneish Thangaraj Selvaraj

**Affiliations:** 1UK Dementia Research Institute, University of Edinburgh, Edinburgh EH16 4SB, UK; a.salzinger@ed.ac.uk (A.S.); vramesh@exseed.ed.ac.uk (V.R.); sparthad@exseed.ed.ac.uk (S.D.S.); siddharthan.chandran@ed.ac.uk (S.C.); 2Centre for Clinical Brain Sciences, University of Edinburgh, Edinburgh EH16 4SB, UK; 3Anne Rowling Regenerative Neurology Clinic (ARRNC), University of Edinburgh, Edinburgh EH16 4SB, UK

**Keywords:** amyotrophic lateral sclerosis, neuronal circuit, synaptic dysfunction, stem cells, spinal cord, motor neurons, cortical neurons, excitability, glutamate excitotoxicity, neuromuscular junction

## Abstract

The primary neural circuit affected in Amyotrophic Lateral Sclerosis (ALS) patients is the corticospinal motor circuit, originating in upper motor neurons (UMNs) in the cerebral motor cortex which descend to synapse with the lower motor neurons (LMNs) in the spinal cord to ultimately innervate the skeletal muscle. Perturbation of these neural circuits and consequent loss of both UMNs and LMNs, leading to muscle wastage and impaired movement, is the key pathophysiology observed. Despite decades of research, we are still lacking in ALS disease-modifying treatments. In this review, we document the current research from patient studies, rodent models, and human stem cell models in understanding the mechanisms of corticomotor circuit dysfunction and its implication in ALS. We summarize the current knowledge about cortical UMN dysfunction and degeneration, altered excitability in LMNs, neuromuscular junction degeneration, and the non-cell autonomous role of glial cells in motor circuit dysfunction in relation to ALS. We further highlight the advances in human stem cell technology to model the complex neural circuitry and how these can aid in future studies to better understand the mechanisms of neural circuit dysfunction underpinning ALS.

## 1. Introduction

Amyotrophic lateral sclerosis is a fatal adult-onset neurodegenerative disorder, characterized by progressive muscle atrophy, paralysis, and eventual death. The disease primarily affects the cortico-spinal motor circuit which originates in the motor cortex and terminates in nerves innervating skeletal muscle (Figure 1), thereby controlling voluntary movement. Specifically, the disease is caused by the loss of both upper motor neurons (UMNs) in the motor cortex and lower motor neurons (LMNs) in the brainstem and spinal cord. UMNs can form direct glutamatergic monosynaptic connections with LMNs in the spinal cord via the corticospinal tract, a feature unique to primates and increased in humans, postulated to correlate with manual dexterity [1,2]. Although neurodegeneration occurs throughout the corticospinal motor circuit, there have been varied hypotheses regarding onset of pathology in ALS, as to whether the origin is in the motor cortex (dying forward) [3,4,5] or spinal cord/muscle (dying back) [6,7,8].

Most ALS cases are sporadic in nature, and approximately 10% of ALS cases are familial. Genome-wide association studies have identified more than 50 genes associated with ALS pathophysiology [9]. Pedigree studies using clinical-based registry studies have ascertained heritability of ALS to be between 40 and 60% dependent on the genetic risk [10,11]. ALS-linked genes can be categorised into three pathways, (1) RNA metabolism, (2) autophagy/protein clearance, and (3) cytoskeletal proteins. The most commonly occurring mutations are within the genes of *C9ORF72*, *SOD1*, *FUS,* and *TARDBP* [12]. Noting that familial and sporadic forms of ALS are indistinguishable both clinically and pathologically, studies performed to understand pathomechanisms have relied predominantly on the genetic forms of ALS. The intronic G_4_C_2_ repeat expansion mutation in the *C9ORF72* gene is the most common gene mutation, observed in approximately 40% of familial ALS and approximately 10% of sporadic ALS patients [9]. Three non-mutually exclusive mechanisms have been postulated by which C9ORF72 mutation leads to cellular dysfunction: (1) intronic G_4_C_2_ repeats is transcribed bi-directionally (sense and anti-sense) and forms intra-nuclear RNA foci which sequesters multiple RNA-binding proteins, (2) Repeat Associated Non-AUG (RAN) translation of G_4_C_2_ repeat RNA to form five different di-peptide repeat proteins (DPRs), and (3) hypermethylation of G_4_C_2_ repeats leading to haploinsufficiency of downstream C9ORF72 [13]. Mutations in the open-reading frame of *SOD1* gene coding for Cu/Zn superoxide dismutase enzyme accounts for approximately 10% of familial ALS cases [14,15]. Despite decades of research, mechanisms of SOD1 related pathology are still not fully understood. Several studies have shown that the gain of toxicity drives disease pathology [16]; however, the role of enzymatic loss-of-function of SOD1 in ALS disease aetiology has also been suggested [17]. While mutations observed in the *TARDBP* gene are rare (<1%), the protein encoded by the *TARDBP* gene—a TDP43 protein—forms cytoplasmic aggregates in approximately 97% of ALS cases including the genetic forms of ALS (exceptions are SOD1-ALS and FUS-ALS), making it hugely relevant for understanding the pathophysiology of ALS [18]. TDP43 is a nuclear-localised RNA/DNA binding protein, with essential functions in the regulation of splicing [19,20,21]. Cytoplasmic TDP43 aggregates leads to both toxic gain of function and the depletion of TDP43 protein from the nucleus, causing a loss of its physiological function [22].

Multiple disease mechanisms have been described in ALS individuals and models of ALS, in both neurons and glial cells. Some key phenotypes include impaired protein homeostasis and RNA metabolism, oxidative stress, glial dysfunction, and changes in neuronal excitability and excitotoxicity [23,24]. A common feature observed along the corticospinal motor circuit is synaptic dysfunction. There are two major groups of synaptic connections which are present on this circuit, the synapses between UMNs and LMNs, and subsequently, LMNs synapse with the muscle via neuromuscular junctions (NMJs). Regulation of synaptic transmission is complex and dependent on various factors, such as modulation of the synaptic vesicle formation, release of the synaptic vesicles into the synaptic cleft, and postsynaptic receptors and subsequent signalling pathways. Neurotransmitters released by the presynaptic neuron can be excitatory or inhibitory [25]. Dysregulation of such proteins (for instance, PSD95, Ubiquilin2, and CHMP2B) may result in altered synaptic structure and function, eventually leading to cognitive and motor impairments, similar to other progressive neurodegenerative disorders [23,26,27,28]. Recent studies have shown that TDP43, C9ORF72, SOD1, and FUS play important roles in maintaining proper synapse development and function [29,30,31], indicating the crucial interplay between molecular and cellular mechanisms in ALS pathology.

ALS and its pathophysiology have been studied in detail over the past few decades using preclinical animal and in vitro models which have been informative in recapitulating aspects of disease pathology [32,33]. However, these models have also translated inadequately to human therapeutics and led to poor clinical trial outcomes [34,35,36]. These constraints have led many groups to turn towards using human pluripotent stem cell (hPSC) derived systems for modelling ALS. Moreover, given the differences in human and rodent corticospinal synaptic connections [37] it is increasingly becoming relevant that human models of ALS are needed to better understand disease mechanisms and translate to more positive therapeutic outcomes. hPSC models are proving important in understanding species-specific recapitulation of disease pathology by ease of genetic manipulation and careful analyses of the temporal and spatial disease profile. Indeed, 2D and 3D hPSC models are emerging as powerful tools to uncover novel mechanisms in many neurodegenerative disorders [38,39,40].

In this review, we summarize the key impairments observed in neuronal activity and synaptic function along the corticospinal motor circuit in ALS patients and in vivo/in vitro models. We discuss some of the crucial unanswered questions and the use of complex hPSC models in understanding ALS disease aetiology for therapeutic intervention. 

## 2. Cortical Neuron Dysfunction in ALS

One of the defining features of ALS pathology is the degeneration of upper motor neurons (UMNs) in the motor cortex and corticospinal tract (CST). Although UMN degeneration and dysfunction in ALS patients is well documented [41], the reasons for this selective vulnerability and the role of the cerebral cortex in aetiology and disease progression are yet poorly understood. 

There have been a few key studies that have examined degeneration and dysfunction of neurons in the cerebral cortex in ALS patients. Diffusion tension imaging and structural MRI has been used to describe disorganization in the CST and reduced precentral cortical ribbon thickness [42]. Cortical thinning in primary motor areas has been widely described [43,44] and correlated with faster disease progression in temporal areas [44]. 

In addition to the loss of neurons, specific hallmarks of degeneration have been reported wherein Betz cells (human UMNs) in both sporadic and familial ALS patients exhibited apical dendrite abnormalities [45,46]. Interestingly, the dendritic abnormalities correlated with a specific loss of PSD95^+^ postsynaptic and pre/postsynaptic colocalized puncta in Betz cells [45]. Reduction in the presynaptic marker synaptophysin has also been reported in the prefrontal cortex of ALS patients and correlated with cognitive decline [47]. Studies have uncovered molecular links to the atrophy pathology wherein atrophy in the motor cortex and CST positively correlated with the TDP43 pathology staging system described by Brettschneider and colleagues [48]. Post-mortem studies from patients describing cortical thinning, dendritic, and axonal degeneration in the cortex and CST has been supported by findings from rodent models of TDP43 (overexpression of cytoplasmic hTDP43 and A315T mutation) and SOD1 (G93A and G86R mutations) [49,50,51,52]. 

A seminal study in 2008 reported cortical hyperexcitability in the motor cortex of pre-symptomatic SOD1 patients using transcranial magnetic stimulation (TMS) prior to the onset of motor symptoms [53]. This study and others have redirected the focus back to the cerebral cortex in ALS disease aetiology and in identifying early neuronal dysfunction. Cortical hyperexcitability, which renders neurons to fire increased action potentials in response to stimuli, has become an increasingly relevant clinical phenotype and a common hallmark observed in both familial and sporadic ALS patients [54,55,56,57,58,59]. TMS is often used to measure cortical hyperexcitability, characterized by features such as reduced motor evoked potentials and increased intracortical facilitation [56]. Other techniques such as fMRI neuroimaging, EEG and MEG recordings have also been used in conjunction with TMS to indicate enhanced network connectivity in the motor cortex and other brain regions in ALS individuals [60,61,62,63,64]. Layer 5 pyramidal neurons in the motor cortex of a hTDP43 mouse model and corticospinal and cortico-cortico projection neurons in the motor cortex of SOD1^G93A^ mice display hyperexcitability [65,66]. In addition, a recent study using SOD1^G86R^ and FUS1^ΔNLS/+^ mouse models described increased susceptibility to pentylenetetrazol, a GABA_A_ receptor antagonist [67] that unravels network hyperexcitability, indicative of cortical network hyperexcitability in these models. 

Given that the observed cortical hyperexcitability could arise from excitation–inhibition imbalance, studies have examined dysfunction in cortical interneurons. TMS on ALS patients has described a possible functional impairment in intracortical inhibitory interneurons [68]. The wobbler mouse model of ALS reported a decrease in GABAergic inhibition [69], and embryonic GAD67^+^ cortical interneuron cultures from SOD1^G93A^ mice showed an attenuation in excitability [70]. SOD1^G93A^ adult mice exhibited PV^+^ layer 5 interneuron hypoexcitability at late pre-symptomatic stages [71] which morphed into a hyperexcitability phenotype [65] at late symptomatic stages, suggesting that changes in interneuron excitability is highly dynamic during disease progression. Examining cortical inhibition could be another key aspect in disease aetiology and understanding the mechanisms underlying the observed pyramidal neuron hyperexcitability.

Interestingly, both TDP43^Q331K^ and SOD1^G93A^ mice displayed increased excitatory synaptic neurotransmission of layer 5 pyramidal neurons at pre-symptomatic stages [72,73]. Moreover, dendritic spine density in a TDP43^A315T^ mouse model has also been described to precede symptom onset [74]. Key recent work on a TDP43 mouse model, wherein cytoplasmic hTDP43 was specifically induced in the cerebral cortex, described an early cortical hyperexcitability phenotype which spread anterogradely through the corticomotor system and led to lumbar LMN degeneration at later stages [66,75]. These data are some of the first to examine the direct consequence of cortical neuron dysfunction on spinal motor neurons in an ALS model. Further, it sheds important light on the interplay between different neurons affected along the corticospinal motor circuit which warrants further research. In summary, data from ALS patients and animal models prompt important questions on understanding the temporal profile of cortical atrophy, with synaptic dysfunction and cortical hyperexcitability possibly preceding UMN loss. 

Human pluripotent stem cell (hPSC) models are valuable tools to answer these key questions in ALS neuronal circuitry and interplay between cortical and spinal motor neurons. hPSC models of the cerebral cortex, particularly 3D models such as organoids, often display multiple cell types (Figure 2), neuronal maturation, synaptic function, network oscillations, and glutamatergic and GABAergic signalling [76,77,78,79,80,81,82]. Such features of hPSC-derived cortical neuron cultures and organoids have established them as robust human model systems to study cortical neuron function, facilitating temporal and spatial analysis and cell-type-specific genetic manipulation of key molecules. 

Although 2D and 3D hPSC models have been widely used in the context of spinal dysfunction in ALS, hPSC-derived cortical neuron models in context of ALS have been very limited to date. Hyperexcitability was observed in a 2D monoculture of human cortical neurons which correlated positively with the expression of the shortened toxic isoform of TDP43 [86]. Another key study showed that cortical hyperexcitability was observed in C9ORF72 cortical neurons, and this was supported with an increase in synaptic density [79]. Given the limited data on human cortical models of ALS, there is yet much to be uncovered on the dynamics of neuronal dysfunction and the underlying mechanisms that drive key cortical neuron phenotypes and their potential implications on the descending corticospinal motor circuit. 

## 3. Altered Excitability in the Lower Motor Neurons in ALS

The spinal cord motor neurons, also known as lower motor neurons (LMNs), are diverse, both in morphology and function and can be classified into α, β, and γ MNs based on the type of muscle fibre they are innervating. Specifically, α MNs innervate extrafusal muscle, while β MNs innervate both extra- and intrafusal muscle, and γ MNs innervate intrafusal muscle only [87]. Of these, α-motor neurons, which are located in the anterior horn of the spinal cord and innervate the force generating extrafusal muscle, are suggested to be selectively vulnerable to degeneration in ALS [88,89]. A single axon from each α MN innervates several extrafusal muscle fibres comprising the motor unit, and depolarization of the neuron causes all its innervated muscle fibres to contract simultaneously [90]. Indeed, one of the major clinical symptoms seen in patients with ALS is the presence of fasciculation or twitching of the muscles [91,92], which may be a feature of altered excitability of the lower motor neurons innervating the muscles [93]. Using high-density surface electromyography (HDSEMG) recordings, a recent study demonstrated the presence of higher frequencies of fasciculations in ALS patients, which could be due to an increase in the excitability of the LMNs innervating the muscles [94]. Indeed, several studies have indicated upregulation of sodium and downregulation of potassium channel conductance in the motor axons with a concomitant change in excitability in patients with both familial and sporadic forms of ALS [95,96,97]. Notably, one of the main mechanisms of action of Riluzole, the only licensed drug for treating ALS, is inhibition of the persistent Na^+^ channels [98].

Furthermore, a study using hPSC-derived motor neurons from patients with SOD1, C9ORF72, or FUS mutations has shown that these neurons are hyperexcitable owing to reduced delayed-rectifier potassium currents [99]. Hyperexcitability in the ALS MNs were rescued by treatment with a KCNQ (Kv7) channel activator, retigabine. Moreover, in a phase 2 clinical trial conducted using retigabine (also known as ezogabine), the excitability of both the upper and lower motor neurons was reduced in a dose-dependent manner by ezogabine, although the treatment window (4 weeks) was too short to observe any alterations in the disease progression [100]. However, recent mechanistic insights have also implicated neuronal hypoexcitability in the ALS pathophysiology. Indeed, in some patients with ALS, the spinal MNs do not exhibit hyperexcitability [101]. Similarly, MNs innervating the fast-contracting fatigable muscle fibres were hypoexcitable in SOD1^G93A^ mice [102]. In addition, several hPSC studies have also shown that motor neurons derived from ALS patients with C9ORF72 [103], FUS, and SOD1 mutations are hypoexcitable due to increased expression levels of voltage-gated potassium channels and decreased levels of sodium channels [104]. It should be noted that neuronal excitability is dynamic, and the biophysical characteristics of the MNs are age-dependent [105]. Devlin and colleagues have shown that hPSC-derived MNs from ALS patients harbouring the C9ORF72 or TARDBP mutations demonstrate an initial hyperexcitability followed by hypoexcitability [106]. Thus, a temporal profile of motor neuronal excitability is key to understanding the pathogenic mechanisms underlying ALS.

The excitability of MNs is mediated by synaptic inputs and disruption of these inputs could lead to aberrant excitability [107]. While synaptic dysfunction has not been studied in detail with respect to the LMNs in ALS, it has been shown that specifically tripartite synapses are lost in post-mortem spinal cord samples from ALS patients harbouring SOD1 and C9ORF72 mutations and in the SOD1^G93A^ mouse model [108]. Furthermore, hPSC-derived neurons from ALS patients showed reduced synapse formation [109] and dendritic arborization of the spinal motor neurons was diminished in transgenic mouse models expressing mutant FUS [110] highlighting the need for studying the molecular mechanisms involved in maintaining the structure and function of the tripartite synapse.

## 4. Glutamate Receptor Dysregulation in ALS

Glutamate is a major excitatory neurotransmitter in the mammalian brain [111,112], and aberrant glutamatergic transmission has been attributed as one of the key mechanisms underlying ALS pathophysiology [91,113]. The presence of excess glutamate in the synaptic cleft may result in an increased calcium influx, thereby resulting in altered excitability and eventual excitotoxicity of the neurons [91,105,113,114]. MNs are selectively vulnerable to glutamate-induced excitotoxicity [115,116,117,118], and the dysregulation of calcium-permeable AMPA (α~-amino-3-hydroxy-5-methyl-4 isoxazole propionic acid) receptors (AMPAR) has been implicated as the underlying mechanism for this selective vulnerability [115,117,119,120,121,122].

AMPARs are tetrameric complexes of four subunits (GluA1-GluA4), which are encoded by four genes, GRIA1-GRIA4 [123,124]. Of these subunits, the GluA2 subunit undergoes constitutive post-transcriptional RNA editing (Q/R editing) and becomes impermeable to Ca^2+^, further distributing the AMPARs into two subclasses: Ca^2+^—permeable (CP) and Ca^2+^—impermeable AMPARs [125,126]. Thus, in addition to expression pattern of AMPAR genes, the subunit composition confers specific molecular and biophysical properties to the neurons and changes during development, learning and memory, and neurological disorders [124,127]. 

Additionally, perturbed Ca^2+^ buffering has also been implicated in making the motor neurons selectively vulnerable to excitotoxicity and death [128]. Specifically, subgroups of motor neurons in the brainstem and spinal cord express low levels of Ca^2+^ buffering proteins, such as calretinin and calbindin [129,130], leading to an overload of Ca^2+^ in the mitochondria of the motor neurons [128,131]. Early studies have shown that cerebrospinal fluid (CSF) from patients with ALS was toxic to cultured neurons [132]. Indeed, CSF from ALS patients when injected into rat pups reduced the number of motor neurons and led to reduced choline acetyl transferase (ChAT) expression [133]. However, studies analysing the levels of glutamate in the CSF have revealed mixed results, whereby some patients had elevated glutamate concentration and other patients had normal glutamate concentrations [134,135], although addition of the AMPAR blockers (CNQX/NBQX) led to increased neuronal survival [132,136]. Thus, whether elevated concentration of glutamate is a prerequisite for excitotoxicity is still up for debate because there is evidence that low glutamate concentration can induce apoptosis in cultured neurons [137].

Similarly, studies using SOD1^G93A^ transgenic mice have shown reduced GluA2 and increased GluA3 levels in the motor neurons [138,139] leading to an overall increase in calcium permeability. Moreover, it was shown that in the spinal motor neurons of patients with ALS, there was a defect in the GluA2 mRNA editing and significant downregulation of the Adenosine deaminase acting on RNA 2 (ADAR2) enzyme [140,141,142] that catalyses the Q/R editing of GluA2 and renders them impermeable to calcium [143]. In the AR2 mouse model, where ADAR2 was knocked out in the motor neurons, the spinal motor neurons degenerated with a concomitant loss of motor function [144]. In motor neurons derived from patients harbouring the C9ORF72 repeat expansion mutation (C9ORF72RE), our group has shown that there was a significant increase in the CP-AMPARs via increase in the GluA1 mRNA expression. Notably, the AMPAR properties were unaltered in human cortical neurons, indicating that this mechanism is specific to motor neurons [121]. Additionally, using post-mortem samples, it was shown that lower motor neurons of patients with sALS and C9ORF72 mutations displayed GluA1 upregulation, while lower motor neurons of those with SOD1 mutations exhibited reduced GluA2 mRNA levels [122]. A presymptomatic SOD1^G93A^ mouse model of ALS has shown higher frequency of excitatory post synaptic currents (EPSCs), indicating an increase in presynaptic glutamate release and an increase in vGlut2 (a vesicular glutamate transporter) expression in the motor cortex [145]. A study examined overexpressed human TDP43^A315T^ in mouse primary pyramidal neurons and found that the neurons exhibited increased levels of GluA1 [146]. Moreover, a TDP43 mouse model (overexpression of cytoplasmic hTDP43) exhibited altered levels of AMPARs in the motor cortex [66]. Interestingly, a study has shown that ADAR2 was downregulated in the spinal motor neurons of patients with sALS, and these cells also showed TDP43 pathology [147] indicating a probable link between TDP43 and GluA2 levels. Thus, results from these studies indicate an underlying AMPAR dysfunction contributing to lower motor neuron degeneration and specific molecular changes, underpinning this need to be explored further.

While AMPAR dysfunction and glutamate excitotoxicity have been largely studied, drug trials emanating from preclinical studies have not fared very successfully [148]. Talampanel, an AMPAR antagonist showed a moderate effect on muscle function and strength in a small phase 2 study but did not show significant effect in a larger trial [149]. Another AMPAR antagonist, perampanel, prolonged motor neuronal survival and function in preclinical mouse model [150] but when used in the clinical trials led to adverse events in ALS patients, such as aggression, somnolence, and dysarthria [151]. These adverse events could be due to perturbed AMPAR levels in other regions of the central nervous system (CNS) by the AMPAR antagonists. Thus, it is imperative to understand the molecular mechanisms underpinning AMPAR dysfunction for generating viable drug targets. Approaches to remove excess glutamate from the synaptic cleft by increasing the expression of the glutamate transporter, EEAT2, using ceftriaxone also did not yield better results [152]. One of the reasons for the failure of these drug trials could be limited bioavailability of the drug in the CNS. Thus, in addition to studying potential drug targets and their mechanism of action, studies should also include analysis of how adequately the candidate drugs can cross the blood–brain barrier to improve therapeutic efficacy.

## 5. Neuromuscular Junction Degeneration in ALS

The neuromuscular junction (NMJ) is the synaptic connection between motor neurons (MNs) and muscle fibres, and degeneration of this synaptic connection occurs early during ALS disease progression as evidenced clinically by altered fasciculation rates and morphology [153,154,155]. Importantly, axonal sprouting of surviving MNs into orphaned muscle was identified as a compensatory mechanism as fasciculation rates momentarily increase, which is associated with muscle reinnervation. Ultimately, fasciculation rates fall, as NMJs and MNs are degenerating. Consistent with patient studies, several ALS animal models have shown NMJ degeneration prior to symptom onset [156] independent of their genotype, confirming that NMJ degeneration is a common pathology across the range of familial and sporadic ALS.

Equally, human in-vitro models have been able to recapitulate NMJ dysfunctions in ALS models [157,158,159,160,161] over a wide range of genotypes, demonstrating that NMJ degeneration is a consistent finding in ALS models. For instance, SOD1^+/G85R^ and PFN1^+/G118V^ human neuromuscular organoids lead to a reduced innervation of NMJs [160]. On the other hand, TDP43^+/G298S^ in organoids and microfluidic cocultures lead to a smaller size of innervated NMJs [160,161].

Considering that 97% of all ALS patients present with cytoplasmic TDP43 mislocalization, understanding the implications of TDP43 pathology on NMJ degeneration is important and has been addressed by several studies [51,162,163,164,165]. Selective expression of cytoplasmic TDP43 under human *NEFH* promoter led to NMJ degeneration prior to significant MN loss [51]. Crucially, when the cytoplasmic expression of TDP43 was reversed after symptom-onset, muscle was re-innervated and motor phenotypes were restored, although the number of MNs remained unchanged. This strongly suggests a compensatory sprouting of remaining motor axons to form new NMJs [51]. Likewise, physiological expression of human mutant form of TDP43 (TDP43^M337V^ and TDP43^Q331K^) led to motor deficits and NMJ degeneration at 9 and 10 months, respectively [163,164]. Furthermore, selective loss of TDP43 in motor neurons resulted in NMJ degeneration and motor deficit [165]. It has been widely known that TDP43 loss-of-function leads to cryptic splicing of a plethora of genes, amongst others Stathmin2, resulting in downregulation of Stathmin2 expression [19,20,21]. Noting that Stathmin2 is a microtubule-associated protein and that genetic evidence in ALS implicates the involvement of cytoskeletal proteins in disease mechanism, it raises the question as to whether Stathmin2 dysregulation through TDP43 LoF could contribute to axon and NMJ degeneration. A recent study by Krus and colleagues [162] showed that NMJ degeneration can be recapitulated by Stathmin2 loss alone. Although Stathmin2 knockout mice exhibit motor impairments and axonal degeneration, no MN loss was observed, even at older age [162]. Overall, these studies suggest that several aspects of the TDP43 pathophysiology are involved in MN and NMJ maintenance and its exact mechanism remains to be discovered. 

Studies have focussed to better understand the mechanisms by which NMJs are degenerating in ALS with the aim to identify drug targets that could slow down NMJ degeneration and/or boost compensatory mechanisms. One such pathway that has been highlighted is local protein synthesis. NMJs are located at the distal end of the MN axon; therefore, local protein synthesis is essential to provide synaptic plasticity and maintenance [166,167]. RNA binding proteins (RBPs), such as TDP43 and FUS together with their corresponding RNA targets, form a membraneless ribonucleoprotein (RNP) complex, which is transported to the distal end of the MN axon and the NMJ for local protein translation. Perturbation of the local translation machinery, such as aggregation of RNPs, has been observed in axonal compartments of mutant FUS MNs [168] as well as at the TDP43 mutant NMJ [169], culminating in reduced functionality of NMJs as measured by muscle activation [169]. Altman and colleagues showed that the clearance of RNP aggregates or restoration of nuclear TDP-43 reversed NMJ phenotype, highlighting NMJ misfunction and degeneration as a consequence of mislocalized aggregates of the RNA binding protein TDP43 [169]. Similarly, mutations of the nuclear-encoded FUS leads to cytoplasmic mislocalisation and formation of insoluble stress granules [170]. FUS mutant MNs, cocultured with healthy myotubes, revealed reduced complexity and number of NMJs [159,171] demonstrating impaired maintenance of mutant FUS NMJs. Additionally, axonal growth of mutant FUS MNs was impaired in both initial outgrowth and regrowth after stress (axotomy) and was rescued by genetic correction of FUS [171]. While the underlying mechanisms leading to impaired NMJ maintenance is not completely understood, it is noteworthy that selective inhibition of HDAC6, known for deacetylating microtubules [172] and restoring axonal transport deficits, ameliorated both the axonal growth level and NMJ pathology [171]. In summary, local protein translation and axonal transport [173] are essential mechanisms of NMJ maintenance and might be affected in ALS [159,169,171,172,173]. Future studies are needed to understand impairments in key pathways and its molecular changes at the NMJ upon and prior to degeneration.

## 6. Glial Contribution towards Neuronal Circuit Dysfunction in ALS

In ALS, MN vulnerability also involves non-cell-autonomous mechanisms owing to impairment in glial cell function [174]. Selective removal of mutant SOD1 in astrocytes [175], microglia [176], and oligodendrocytes [177] slowed disease progression. Better understanding the mechanisms by which glial cells contribute to ALS pathogenesis could pave the way for novel therapeutic targets. 

Astrocytes modulate synaptic transmission by expressing glutamate transporters such as EAAT1 and EAAT2 that take up excess glutamate in the synaptic cleft and can be dysregulated during neurological disorders [178,179]. Earlier studies from Rothstein and colleagues observed selective downregulation of astrocytic EAAT2 both in the motor cortex and spinal cord of ALS [180], and subsequent studies knocking down GLT-1 in astrocytes of organotypic spinal cord slice cultures resulted in toxicity to motor neurons. Crucially, motor neuron toxicity through EAAT2 knockdown in astrocytes was prevented by adding an AMPA/kainate receptor antagonist [181]. These seminal studies suggest that impaired clearance of glutamate from synaptic cleft by ALS astrocytes—owing to reduced EAAT2 transporters—causes excitotoxicity in motor neurons. Astrocytes are also known to modulate expression of AMPA receptors in the neurons [182], and a recent study highlighted that astrocytes carrying FUS mutation induce toxicity to motor neurons by upregulation of Ca^2+^-permeable AMPA receptor, GluA1, thus rendering motor neurons susceptible to excitotoxicity [183]. Additionally, conditioned media from SOD1 mouse astrocytes was specifically toxic to the motor neurons, suggestive of soluble neurotoxic factors released by the astrocytes in ALS [184,185].

Human stem cell disease modelling has enabled the mechanistical delineation of the non-cell autonomous contribution of glial cells in ALS. hPSC-derived astrocytes from patients harbouring C9ORF72 mutations did not affect motor neuronal viability but led to hypo-excitability of the motor neurons by reducing the sodium and potassium currents [186]. Altered potassium homeostasis in the synaptic cleft, which is primarily regulated by the astrocytic inwardly rectifying potassium (Kir) channels [187], can contribute to physiological dysfunction in neurons [188]. Specifically, expression of the Kir4.1 channel is downregulated in SOD1 mouse model and hPSC-derived astrocytes. Selective loss of Kir4.1 in astrocytes led to altered fast-fatigable αMNs (those that are vulnerable in ALS) size, function and led to reduced peak strength without overtly affecting the survival of motor neurons [189]. Moreover, ALS astrocytes have also been shown to contribute to axonal damage/degeneration [190]. Human hPSC-derived astrocytes carrying FUS mutation when cocultured with motor neurons and myotubes resulted in toxic effect on neurite outgrowth and impaired NMJ formation and function by modulating WNT/β-catenin pathway on motor neurons [191]. There is an emerging concept of microglia, a CNS-resident immune cell regulating neuroinflammation, affecting neuronal physiology in the disease context. Indeed, cell-intrinsic immune dysfunction in microglia harbouring the C9ORF72 mutation confers increased vulnerability to excitotoxicity to both healthy and C9ORF72 motor neurons [192,193]. In summary, these studies highlight that glial cells are key determinants of ALS pathogenesis, and it is necessary to precisely understand the mechanisms by which glial cells induce neuronal physiological toxicity [192,193]. In summary, these studies highlight that glial cells are key determinants of ALS pathogenesis, and it is necessary to precisely understand the mechanisms by which glial cells induce neuronal physiological toxicity.

## 7. Use of Human Stem Cell Models to Study Neuronal Circuit Dysfunction in ALS

In the last few decades, important advances have been made in understanding the pathophysiology and neuronal circuit dysfunction in ALS. However, given the largely unsuccessful number of clinical trials [194], there is an increasing need for more complex, disease-relevant models. Moreover, studies showing species differences within the corticospinal motor circuit, highlights the importance of complementary human models [195,196,197], to facilitate a better understanding of both sporadic and familial ALS pathophysiology. The emerging field of hPSC models has proven to be a robust preclinical model of neurodegenerative disorders such as Parkinson’s, Alzheimer’s, Amyotrophic lateral sclerosis, and Huntington’s disease by recapitulating key aspects of human pathology [198,199].

While 2D hPSC models provide valuable insights into cellular and molecular mechanisms of disease, emerging 3D organoid models (Figure 2) can offer additional benefits, such as the recapitulation of cellular interplay and the development of complex neural circuitry. In addition, key questions around cellular autonomy in pathophysiology can be addressed by the presence of multiple neural cell types (such as progenitors, subtypes of neurons, astrocytes, and oligodendrocytes) in organoid models [200,201]. Recent seminal studies by Pasca and others have described assembloids that combine brain, spinal cord, and muscle organoids to generate 3D systems of the corticospinal motor circuit [202,203] (Figure 2). Such models, which develop complex neuronal circuits, have revolutionized the organoid field, facilitating the study of circuit disorders such as ALS. Nevertheless, organoid models do have several caveats and limitations which need to be considered in experiment design. Difficulties arise in long-term cultures as necrotic cores, for instance, are a common occurrence in organoids [204]. Furthermore, most organoid protocols still lack vasculature and certain cell types, e.g., microglia, but first studies have successfully implemented these [205,206,207].

In the context of ALS, the benefits of hPSC models lie within key areas ranging from gene-editing, as well as drug discovery and development of high-throughput screening platforms. Furthermore, such systems are especially important in the context of studying neuronal circuit dysfunction given the key differences in the human and rodent CST and NMJ [195,196,197], which cannot always be faithfully recapitulated in animal models. Key questions in understanding ALS circuit dysfunction could be addressed using organoid/assembloid models to study long-range neuronal circuits [202,203,208,209,210], synapses [160,211,212], and myelination [213].

In summary, combining the ease of genetic manipulation of known ALS-linked genes and the use of patient-derived or sporadic stem cells, with the complexity of human organoid models, will allow the field to truly recapitulate ALS disease development and answer key mechanistic and aetiological questions as well as aid in the development of novel therapeutics. In addition to understanding the impact of ALS-causing pathogenic mutations, these models also enable us to study the impact of region-specific neurons (upper or lower motor neurons) and glia on circuit dysfunction. Combining these models with the latest ‘omics’ technologies such as spatial transcriptomics and single-cell proteomics and transcriptomics will provide further insights in novel therapeutic targets and potential biomarkers.

## Figures and Tables

**Figure 1 cells-13-00792-f001:**
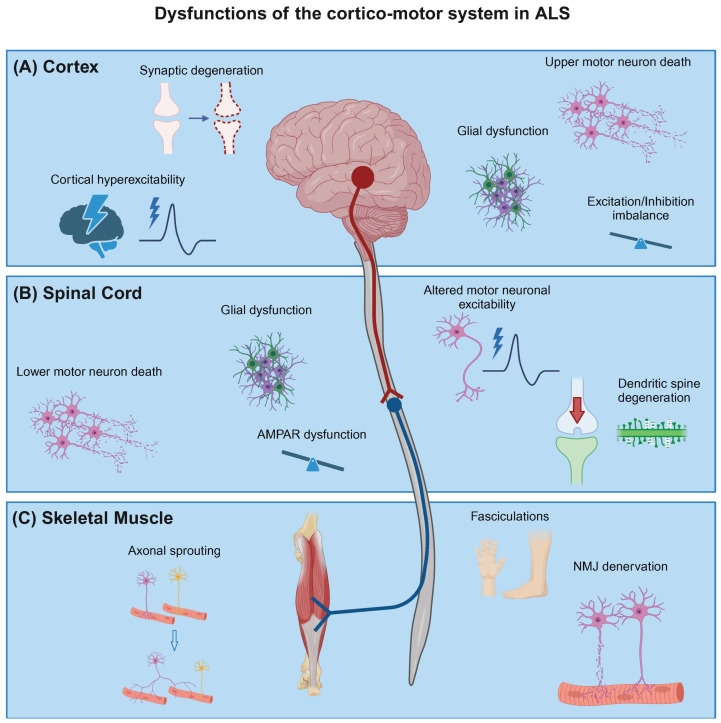
Schematic of dysfunctions of the cortico-motor system in amyotrophic lateral sclerosis. (**A**) UMNs in the motor cortex synapse with LMNs in the spinal cord via the corticospinal tract. This circuit degenerates in ALS with patients exhibiting loss of cortical neurons and dendritic and synaptic degeneration. Moreover, changes in neuronal physiology have been observed such as cortical hyperexcitability and reduced cortical inhibition. (**B**) LMNs in the anterior horn of the spinal cord are particularly vulnerable to degeneration in ALS. Further, altered excitability, dysregulated AMPAR subunit expression, glutamate-mediated excitotoxicity, and degeneration of dendritic spines have been observed in these LMNs. In addition, astrocytes, oligodendrocytes, and microglia (glial cells) undergo functional changes in ALS. (**C**) LMNs connect to the skeletal muscle via the neuromuscular junction (NMJ) which is denervated early during the disease progression. Initially, surviving LMNs reinnervate orphaned muscle by compensatory axonal sprouting, clinically evidenced by altered fasciculations. Ultimately, muscle fibres are fully denervated. Created with BioRender.com.

**Figure 2 cells-13-00792-f002:**
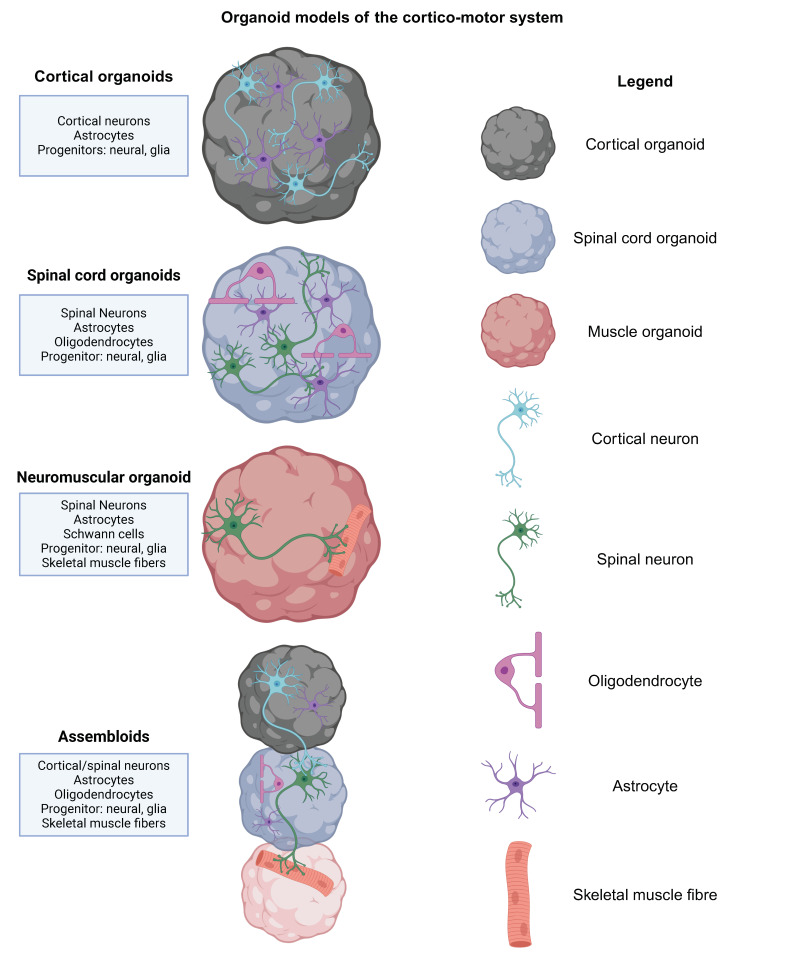
Organoid models of the cortico-motor system. The last decade has seen the emergence of several organoid models of the corticomotor circuit. Human stem cell-derived organoid systems can be generated to model certain aspects of the cerebral cortex, spinal cord, and muscle. Most 3D models generate the primary neural cell types, neuronal and glial progenitor, excitatory and inhibitory neurons, astrocytes and oligodendrocytes. Neuromuscular organoids may contain schwann cells, which are essential for neuromuscular junction maintenance. The generation of assembloid models has progressed the field allowing the study of complex neuronal circuits by assembling region-specific organoids. With increasing interest, researchers aim to incorporate other resident CNS cells such as microglia and vasculature in organoids and, more recently, in assembloids [83,84,85]. Created with BioRender.com.

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
