# Peer review of "Neuronal Circuit Dysfunction in Amyotrophic Lateral Sclerosis"

_cells, 2024, doi:10.3390/cells13100792_

Round 1

Reviewer 1 Report

Comments and Suggestions for Authors

In their manuscript Salzinger et al. review current insight into neuronal circuit dysfunction in ALS. The manuscript is overall well written and covers a number of aspect relevant to circuit function. The topic is rather novel and of high interest, however a number of very similar reviews exist already, which were not cited (e.g. Mora et al., 2023, Gunes et al., 2020). I recommend to refer to these and highlight the novelty of the current review to emphasize the advance in the field.  An additional figure or table  summarizing details on the described findings, e.g. AMPA dysregulation or diverse types of neurons involved and their changes in ALS patients, hPSC and mouse models etc would be great.

I came across a number of language, terminology or overall structure aspects, which I would like to share with the authors:

-          Half of the abstract is focused on parts of the motor circuitry in general. This is to my mind unnecessary as it is a) incomplete (other critical structures like the basal ganglia are not listed) and b) the mentioned cerebellum is not subject of this review etc.

-          Intro: page 1 – ‘muscle nerve endings’ – should probably say “nerve endings targeting muscles” or similar

-          Fig 1 legend: ‘NMJs are denervated’ – it is probably more accurate to say the target – so muscle fibers  - are denervated

-          The reference Mejzini et al. 2019 occurs twice as a and b – but there is only one in the Reference list

-          Line 55: ‘risk genes’  - given the high penetrance for most of these mutated genes I recommend to call those ALS-linked genes and not risk genes

-          Line 74-76: I believe there is good evidence arguing primarily for a toxic gain of function of SOD mutations and to a much lesser degree for a loss of function. Please at least cite supporting references if you make the claim it is not solved yet.

-          Line 85-86: please slightly change this sentence –“ …described in ALS individuals and other models of ALS,…” as ALS patients are no models

-          Line 93-94: “..release of the synaptic vesicles into the synapse..” – release into the synaptic cleft is probably more accurate

-          Line 96 – 97: “Dysregulation of these proteins…” – which proteins are meant here? Please specify

-          Line 103 – 117: the focus here on hPSC models diverts from the main focus of this review and probably is better suited for a future outlook or methodological part?

-          Line 124: “..degeneration of the motor cortex…” – this is not entirely true as only particular cells are degenerating not the entire motor cortex

-          Line 129 – 130: “There have been a few key studies that have examined degeneration and dysfunction of specific populations of cortical neurons in ALS patients.” The subsequent sentences and references however don’t really address individual neuronal populations (with the exception of Betz cells) – cortical thinning, diffusion tensor imaging or overall alterations in synaptic density do not allow to pinpoint individual neuronal populations or cell types. This part rather addressed alterations found in (motor) cortex in general.

-          Line 156 – 158: “Layer 5 pyramidal neurons in the motor cortex of TDP43 (over- 156 expression of cytoplasmic hTDP43) mouse model and corticospinal and cortico-cortico projection neurons in the motor cortex of SOD1G93A mice exhibit cortical hyperexcitability (Dyer et al., 2021a; Kim et al., 2017).” – individual neurons can display hyperexcitability and probably contribute to the phenomenon of cortical hyperexcitability – but neurons themselves do not display ‘cortical hyperexcitability’

-          Line 160 – 161: “… cortical network hyperexcitability in response to pentylenetetrazol, a GABAA receptor agonist..” – this is a GABAA antagonist and the network hyperexcitability is apparently present in these models (see Scekic-Zahirovic, 2021) and was demonstrated or unmasked upon PTZ application as it causes epileptic discharges.

-          Line 184 – 200 : see above, the detailed description of organoids doesn’t fir there especially when afterwards jumping back to C9 cortical neurons? There are often jumps between mouse models, hPSC and human patient findings. I would recommend to structure it better (e.g. there is also clearly more work done using TMS to assess cortical inhibition and excitation in humans)

-          Line 208: “..based on their muscle target..” – rather the type of muscle fiber is critical here not the muscle itself

-          Line 210: similar – it is the “extrafusal muscle fiber” that is targeted not the entire muscle

-          Line 245: “..spatiotemporal profile..” – not sure about the spatial aspect here which is not mentioned before unless this refers to UMN vs LMN?

-          Please also list and discuss the work by Daniel Zytnicki and colleagues who have tried to answer the question of LMN excitability changes by performing in vivo patch clamping with quite interesting results

-          Line 323: “..shown glutamate accumulating in ..” –  the paper demonstrates an increase of vGlut2 expression but as far as I can tell no direct evidence of glutamate accumulation

-          Line 354 - : please also add earlier work on the potential secretion of toxic factors by astrocytes

-          Line 403: I would change the order slightly and add the para on NMJs after LMN are discussed and before glia are described. I would also check the language here. NMJs are the synapses so technically they are not innervated – hence cannot be denervated. It may be semantics but I personally would rather say NMJ degeneration maybe (the muscle however becomes denervated)

-          Could you authors comment on wether there are attempts to add glia cells the corticospinal tract assembloids? This is somehow mentioned in the figure caption but now references are cited. I thus recommend to expand this part in the text.

-          Please also discuss the shortcomings of such organoid models and potential pitfulls.

-          Fig 2 could be improved as the main aspect which are the cells in these ‘clumps’ are barely visible

Comments on the Quality of English Language

Minor language changes.

Reviewer 2 Report

Comments and Suggestions for Authors

In this manuscript, the authors provide a comprehensive summary of the neuronal dysfunctions and synaptic impairments observed within the corticospinal motor circuitry of ALS patients, as well as in corresponding in vivo and in vitro models. The potential role of human stem cells in ALS research is thoroughly explored, offering valuable insights into the mechanisms underlying ALS disease models and highlighting recent advancements in ALS therapeutics.

Major comments:

1.     The abstract outlines three neuronal circuit modules involved in motor control, yet it omits the discussion of cerebellar motor impairment associated with ALS—a topic not addressed within the main body of the article. Furthermore, the abstract falls short in encapsulating the manuscript's full scope and content.
2.     Several sections require refinement for clarity and relevance, particularly the descriptions of glutamate receptors (lines 259-276), the introductory background on glial cells (lines 355-379), and the methodologies employed in generating human stem cell-derived 3D neuromuscular junction (NMJ) organoid models (lines 499-516). The content in these sections often strays from direct ALS relevance, and judicious pruning is advised to maintain focus on the disease.
3.     I recommend consolidating sections 7 and 8, which describe NMJ degeneration as a hallmark phenotype of ALS and the application of human stem cells in studying NMJ function in relation to ALS. These sections could be more effectively presented under a single heading, such as "Emerging Models for ALS Study."
4.     While the application of stem cells is a recurring theme throughout the manuscript, the final two sections dedicated to this topic lack a cohesive generalization. The authors should aim to synthesize these discussions to present a more unified perspective on the role of stem cell research in ALS.

Minor comments:

1.       In line 36, add more reference to prove the correctness of this description.

 2.       In line 74, the reference was old. Was there latest reference to count the proportion of SOD1 in ALS patients.

 3.       Figure 2 has no citation in the review.

 4.       In line 326, This paragraph should not be a separate paragraph, it should be aggregated into the previous paragraph.

Comments on the Quality of English Language

No.

Reviewer 3 Report

Comments and Suggestions for Authors

In this review manuscript, the authors provided comprehensive overview of the current understanding of neuronal circuit dysfunction in amyotrophic lateral sclerosis (ALS). Briefly, the review covered a wide range of relevant topics, including cortical neuron dysfunction, altered excitability in lower motor neurons, glutamate receptor dysregulation, glial contribution, neuromuscular junction denervation, and the use of human stem cell models in ALS research. This review manuscript is well-written and provides valuable insights into the current understanding of neuronal circuit dysfunction in ALS and the potential of human stem cell models in advancing ALS research. With minor revisions to address the suggested points, this manuscript would make a great contribution to the field.

Specific Comments:

1. The authors could further expand the discussion on the potential interplay between the pathological changes observed at different levels of the corticospinal motor circuit. For instance, how does the cortical hyperexcitability influence the function and survival of spinal motor neurons?

2. The authors may want to explore the potential implications of the reviewed findings for the development of novel therapeutic interventions. 
